# Both Feline Coronavirus Serotypes 1 and 2 Infected Domestic Cats Develop Cross-Reactive Antibodies to SARS-CoV-2 Receptor Binding Domain: Its Implication to Pan-CoV Vaccine Development

**DOI:** 10.3390/v15040914

**Published:** 2023-03-31

**Authors:** Janet K. Yamamoto, Lekshmi K. Edison, Dawne K. Rowe-Haas, Tomomi Takano, Chen Gilor, Chiquitha D. Crews, Apichai Tuanyok, Ananta P. Arukha, Sayaka Shiomitsu, Heather D. S. Walden, Tsutomu Hohdatsu, Stephen M. Tompkins, John G. Morris Jr., Bikash Sahay, Subhashinie Kariyawasam

**Affiliations:** 1Department of Comparative, Diagnostic, and Population Medicine (CDPM), College of Veterinary Medicine, University of Florida, Gainesville, FL 32610, USA; 2Laboratories of Comparative Immunology & Virology for Companion Animals, CDPM, College of Veterinary Medicine, University of Florida, Gainesville, FL 32610, USA; 3Center for Vaccines and Immunology, College of Veterinary Medicine, University of Georgia, Athens, GA 30602, USA; 4Laboratory of Veterinary Infectious Disease, Department of Veterinary Medicine, Kitasato University, Tokyo 108-8641, Japan; 5Department of Small Animal Clinical Science, College of Veterinary Medicine, University of Florida, Gainesville, FL 32610, USA; 6Department of Infectious Diseases and Immunology, College of Veterinary Medicine, University of Florida, Gainesville, FL 32610, USA; 7Emerging Pathogens Institute, University of Florida, Gainesville, FL 32610, USA

**Keywords:** SARS-CoV-2, feline coronavirus, receptor binding domain, cross-reactivity

## Abstract

The current study was initiated when our specific-pathogen-free laboratory toms developed unexpectedly high levels of cross-reactive antibodies to human SARS-CoV-2 (SCoV2) receptor binding domain (RBD) upon mating with feline coronavirus (FCoV)-positive queens. Multi-sequence alignment analyses of SCoV2 Wuhan RBD and four strains each from FCoV serotypes 1 and 2 (FCoV1 and FCoV2) demonstrated an amino acid sequence identity of 11.5% and a similarity of 31.8% with FCoV1 RBD (12.2% identity and 36.5% similarity for FCoV2 RBD). The sera from toms and queens cross-reacted with SCoV2 RBD and reacted with FCoV1 RBD and FCoV2 spike-2, nucleocapsid, and membrane proteins, but not with FCoV2 RBD. Thus, the queens and toms were infected with FCoV1. Additionally, the plasma from six FCoV2-inoculated cats reacted with FCoV2 and SCoV2 RBDs, but not with FCoV1 RBD. Hence, the sera from both FCoV1-infected cats and FCoV2-infected cats developed cross-reactive antibodies to SCoV2 RBD. Furthermore, eight group-housed laboratory cats had a range of serum cross-reactivity to SCoV2 RBD even 15 months later. Such cross-reactivity was also observed in FCoV1-positive group-housed pet cats. The SCoV2 RBD at a high non-toxic dose and FCoV2 RBD at a 60–400-fold lower dose blocked the in vitro FCoV2 infection, demonstrating their close structural conformations essential as vaccine immunogens. Remarkably, such cross-reactivity was also detected by the peripheral blood mononuclear cells of FCoV1-infected cats. The broad cross-reactivity between human and feline RBDs provides essential insights into developing a pan-CoV vaccine.

## 1. Introduction

In a retroactive serological study, our laboratory discovered that specific-pathogen-free (SPF) toms developed minor episode(s) of diarrhea with only low production of FCoV antibodies (Abs) but unexpectedly high levels of cross-reactive Abs to human SCoV2 RBD. Diarrhea began shortly after the initial mating with feline coronavirus (FCoV)-positive queens on 4 December 2019 and continued even after separation from their queens on 10 January 2020. The current work is based on the limited sera collected during COVID-19. More importantly, we were alerted on 7 August 2020 by our collaborators at the University of Georgia (UGA) that our “specific-pathogen-free (SPF)” kittens were infected with either SARS-CoV-2 (SCoV2) or feline coronavirus (FCoV). The goals of our current studies: (i) confirm the original observation, (ii) determine if such SCoV2 cross-reactivity is observed with the sera from laboratory cats infected with either FCoV serotypes 1 or 2 (FCoV1 and FCoV2), and (iii) identify, produce, and use the FCoV RBDs to differentiate FCoV1-infected cats from FCoV2-infected cats serologically and to achieve our first two goals. The findings from current studies should lead to future studies combining SCoV2 RBD and FCoV RBDs as pan-CoV vaccine immunogens against FCoV and SCoV2 infections in cats.

The global prevalence of FCoV infection ranges from 6.6–95% in multi-cat households and catteries [1,2,3]. FCoV is distributed into two phylogenic lineages of FCoV1 and FCoV2. Both serotype FCoVs infect predominantly epithelial cells of the gastrointestinal tract and cause mild gastrointestinal disease (e.g., diarrhea, vomiting, and transient weight loss) in domestic cats, especially in kittens. Often, upon chronic infection, these viruses can also mutate into pathogenic and fatal variants. These variants, called feline infectious peritonitis viruses (FIPVs), infect monocytes and macrophages, spreading throughout the body [4,5,6]. No known cases of FCoV infection in humans have been reported. However, the FCoV sequence sections have been found in recombinant coronaviruses infecting humans [7,8,9]. In comparison, many cases of SARS-CoV-2 (SCoV2) infection in pet cats have been reported worldwide through transmission from COVID-19-positive owners to their pet cats. These cats displayed mild symptoms or remained asymptomatic, even when found positive for SCoV2 by RT-PCR [10,11,12]. Experimental inoculation of laboratory cats with a human isolate of SCoV2 caused the infection, ranging from mild to asymptomatic symptoms. These infected cats were diagnosed as RT-PCR positive through nasal and/or fecal swabs [13,14,15]. SCoV2 transmission from inoculated cats to non-inoculated cats through contact exposure has been demonstrated by multiple research groups worldwide [13,15]. Similarly, transmissions of SCoV2 from COVID-19-positive owners to their symptomatic and asymptomatic pet dogs have been reported [11,12,16]. Experimental infection of laboratory dogs also confirmed that dogs could be infected with SCoV2 [13,14].

Studies have shown that FCoV-infected cats develop antibodies that cross-react with SARS-CoV1 (SCoV1) nucleocapsid (NC) [17] but not with SCoV1 spike 1 (S1) glycoprotein [18], where the SCoV1 receptor binding region (RBD) is located [19]. SCoV1 infection in humans was first discovered in November 2002 in a patient with atypical pneumonia in Guangdong, China [20]. By February 2003, this virus was rapidly transmitted to a large population in Hong Kong, causing severe acute respiratory syndrome (SARS). The intermediate host for SCoV1 was suspected to be a masked palm civet [21]. SCoV1 preparation isolated from a patient was inoculated intratracheally into six laboratory cats. This resulted in an asymptomatic infection in all four cats, based on positive virus isolation and viral RT-PCR of the pharynx, trachea, and lungs [22]. Furthermore, two non-inoculated cats housed together with the SCoV1-inoculated cats became PCR positive starting day 2 of contact exposure with a peak titer at day 6. By day 28, they had seroconverted with a virus-neutralizing antibody (NAb) titer of 40 and 160. One pet cat living in an apartment block with over 100 residents who were positive for SCoV1 was also positive for SCoV1 infection [22].

Both SCoV1 and SCoV2 belong to the genus Betacoronavirus of the family Coronaviridae, while FCoV1 and FCoV2 belong to the genus Alphacoronavirus based on phylogenetic analyses [23]. Both SCoV1 and SCoV2 RBDs bind with human angiotensin-converting enzyme-2 (hACE2) to infect hACE2-expressing human cells and serve as the major target of the NAbs generated by the infected hosts [24]. In contrast, neither FCoV1 nor FCoV2 uses ACE2 as their host cell receptor [25,26]. The cell receptors for FCoV2 and canine coronavirus serotype 2 (CCoV2) are reported to be feline and canine aminopeptidase-N (fAPN and cAPN), respectively. This is similar to the human cold Alphacoronavirus (HCCoV) 229E, using human APN (hAPN) as its primary cell receptor [25,27,28]. The cell receptor for FCoV1 is still unknown, and this is complicated by the fact that FCoV1 isolates do not readily replicate in cell cultures [6,29]. FCoV1 infection is more prevalent worldwide, including in the US, while the FCoV2 infection is found predominantly in Southeast Asia [1,2,3,6,30]. FCoV2 is reported to be a recombinant FCoV1 backbone with CCoV2 spike (S) glycoprotein [29,31]. Since hACE2 is the major cell receptor for SCoV2, SCoV2 infections of cats and dogs are reported to be mediated by the feline ACE2 (fACE2) and canine ACE2 (cACE2), respectively. This finding is based on their amino acid (aa) sequence similarity of fACE2 and cACE2 to hACE2 and the binding analyses of SCoV2 to species-specific ACE2 [23,24,32,33].

The aa sequence comparison between SCoV1 and SCoV2 S proteins demonstrates a considerably high sequence identity of 76%, and their RBDs similarly show a high sequence identity of 73% [34,35,36]. Hence, the aa sequence similarity should be much higher. The sera from FCoV1- and FCoV2-infected cats cross-react with SCoV1 NC but not with SCoV1 S1 glycoprotein [17,18]. Thus, it can be hypothesized from those reports that sera from FCoV1- and FCoV2-infected cats can cross-react with SCoV2 nucleocapsid but not with SCoV2 S1 glycoprotein. Consequently, the sera do not cross-react with the SCoV2 RBD, which resides on the S1 glycoprotein. Furthermore, a few aa sequence analysis studies demonstrate that the Wuhan SCoV2 S glycoprotein is distinctly different from the S glycoproteins of FCoV1 and FCoV2 [23,31]. The location for FCoV1 and FCoV2 RBDs has yet to be determined by biological analysis. However, it has recently been predicted to be around residues 526–676, based on RBD sequence locations of porcine enteric diarrhea virus (PEDV) RBD at B residues 510–640 and transmissible gastrointestinal enteric virus (TGEV) RBD at D3 residues 500–651 [37,38,39]. The TGEV RBD has been reported to bind to the porcine APN as its primary host cell receptor [27,39], but whether PEDV RBD binds to pAPN as its host cell receptor is still controversial [38,40]. The RBD sequence prediction of both PEDV and TGEV was based on monoclonal Ab (MAb) studies, identifying the most potent neutralizing MAb(s) to these porcine Alphacoronaviruses reacting to B and D3 residue regions, respectively [38,39]. Although neutralizing MAb (nMAb) studies have been performed against FCoV2 [41,42], the location and native conformation of FCoV1 and FCoV2 RBDs have yet to be clearly defined and confirmed by the biological analyses required for use as vaccine immunogens.

Our original observation was extremely unexpected based on publications available in 2020–2021, as described above. The feces from the UGA queens, the mothers of these kittens, were tested in September 2020 for FCoV by an RT-PCR method commonly used for feces detection [43]. At that time, the results were negative. Our queens were kindly donated to us by UGA on 24 October 2019, before the start of the COVID-19 pandemic in early 2020. As confirmation or to improve our understanding of the original finding, these sera from FCoV2-infected cats were tested for their cross-reactivity to SCoV2 RBD by a stringent ELISA, immunoblot analyses, SCoV2 RBD blocking assay, and SCoV2 RBD stimulation of peripheral blood mononuclear cells (PBMC). Lastly, we further characterized the nature of FCoV2 transmission in cats by inducing cross-reacting Abs to SCoV2 RBD.

## 2. Materials and Methods

### 2.1. Human and Animal Populations

The single study using blood from two human subjects (Appendix A) has been performed under our approved University of Florida Institutional Review Board IRB202002902. All other studies used laboratory cats that were bred and cared for under the UF IACUC protocols 201801838 through 202001838, and the sera from pet cats collected under IACUC protocol 201803990. Drs. T. Takano and T. Hohdatsu kindly provided the plasma from Japanese laboratory cats inoculated orally/nasally with either FcoV1 UK-2 or FcoV2 79-1146 (0.8-mL oral and 0.2-mL nasal; 1-mL/cat of 10^5^ TCID_50_) and collected at 60 days post-inoculation. The SPF toms were inbred from SPF cats initially derived from intact females (Harlan Sprague Dawley, Inc., Indianapolis, IN, USA) and male SPF cats (Cedar River Laboratories, Mason City, IA, USA). Dr. Andrew R. Moorhead of UGA donated four queens (UGAQ1, UGAQ2, UGAQ3, and UGAQ4) in the fall of 2019, which were initially purchased by UGA from Liberty Research Inc (Waverly, NY, USA). Juvenile cats were generated by mating the donated queens with the three SPF toms at a UF laboratory. The eight group-housed laboratory cats were initially purchased from Liberty Research, Inc. Their animal code did not show the same lineages as the UGA queens and were considered different cat lineages, unrelated to both UGA queens and UF toms. The second blood collection was obtained 15 months later, except for cat G-5, which was euthanized shortly after the first blood collection for unrelated medical reasons. All cats purchased from Liberty Research, Inc. were vaccinated against rabies using the RABVAC3 vaccine and vaccinated against feline panleukopenia, calici, and rhinotracheitis viruses, and against the hemorrhagic feline calicivirus strain using the Fel-O-Vax PCT + Calicivax vaccine. Our animal workers, including the animal care service workers, were confirmed negative by RT-PCR for SCoV2 and COVID-19 when working with our cats.

### 2.2. SARS-CoV-2 RBD Peptides

Two different versions of SCoV2 were used: the University of Florida-RBD (UF-RBD) and MassBiologics-RBD (MB-RBD). The UF-RBD was produced using Harvard Wuhan RBD plasmid kindly provided by Dr. Aaron G. Schmidt (Ragon Institute) and expressed in EXPI293F cells. UF-RBD had an HRV cleavage site, an 8× histidine tag, and a streptavidin-binding peptide tag to assist in the purification of the RBD [44]. The MB-RBD was kindly provided by MassBiologics (Boston, MA, USA) under a UF/MB MTA and constructed from pcDNA with c-Myc (EQKLISEEDL) and 6× His tags [45].

### 2.3. Feline Cell Lines

Crandell feline kidney (CrFK) fibroblast, *Felis catus* 9 (Fc9), and *Felis catus* whole fetus-4 (Fcwf-4) cells were provided by Dr. Niels Pederson of the University of California, Davis. All of these feline cell lines were maintained on Eagle MEM media (Cat. #10-009-CV, Mediatech Inc., Manassas, VA, USA) supplemented with 10% fetal bovine serum (FBS) and 50 μg/mL gentamycin. These cells were maintained at 37 °C with 5% CO_2_ and passaged every 2–3 days.

### 2.4. Production and Partial Purification of FCoV2 Whole Virus

The CrFK cells were infected with FCoV2 79-1146 (ATCC, Manassas, VA, USA) to produce the stock of crude FCoV2 inoculum for in vitro infection studies and partially purified FCoV2 whole-virus (WV) stock. The infected fluids collected directly from the culture flask (175 cm) were pooled and clarified free of cell debris by low-speed centrifugation at 2800–3000 rpm for 45 min at 5 °C. In addition, the freeze-thawed culture fluids from the frozen flasks with residual media were pooled and then clarified free of cell debris by low-speed centrifugation. They were then combined with a portion of the clarified culture fluid from above, from one part of clarified direct culture fluid to two parts of clarified cell-debris fluid. The following two methods partially purified both direct and combined clarified fluids. In the first method, the combined clarified fluid was directly concentrated to 10–50 fold by Vivaspin 20 Centrifugal Concentrators (Sartorius, Gottingen, Germany) with a PES membrane of 100 k MW cut off. The concentrated virus fluid underwent an additional 4–6 washes with PBS using the centrifugal concentrator until the phenol red from the culture media was faintly pink in color. This method resulted in a preparation that had a reasonably high FCoV2 load (1.2 mg/mL) with detectable levels of 180 kDa S glycoprotein but almost undetectable BSA (67 kDa). This preparation was used for developing immunoblot strips. The second method consisted of a concentration of direct-clarified fluid using multiple centrifugal concentrators with minimal PBS washes. As a result, this preparation retained the phenol red color and about 5% residual BSA from FBS. This method provided a virus preparation at a high concentration (150 mg/mL) needed for use in screening a large number of cat sera with FCoV2-WV ELISA.

### 2.5. Transfection and Expression of Expi293F Cells with RBD Plasmids and Purification of RBD Proteins

The plasmid pVRC containing human codon-optimized RBD constructs were transiently transfected into Expi293F cells using the ExpiFectamine™ 293 Transfection Kit (Thermo Fisher, Waltham, MA, USA). Briefly, the cell density was adjusted to 3 × 10^6^ cells/mL in a final volume of 100 mL of Expi293 expression media and allowed to grow 24 h to reach a final density of 5.5 × 10^6^ cells/mL. The plasmid DNA (1 µg/mL) and ExpiFectamine^TM^ 293 reagent were individually diluted with Opti-MEM Reduced Serum Medium (Thermo Fisher, Waltham, MA, USA), incubated 5 min at room temperature (RT), and then mixed together. The ExpiFectamine 293/plasmid DNA mixture was incubated at RT for 20 min and mixed with Expi293F cells. The cells were incubated on an orbital shaker in a 37 °C incubator with 8% CO_2_. After 24 h of incubation, transfection enhancers-1 and 2 were added and incubated for three days. Then, the cell culture was centrifuged at 1800× *g* for 30 min to collect the supernatant for protein purification. The culture supernatant was concentrated to a final volume of 5 mL using a Macrosep Omega Advance Centrifugal Device with a cutoff of 10 kDa (PALL Laboratory, Port Washington, NY, USA). The concentrated supernatant was passed through an equilibrated TALON Metal Affinity Resin (Takara Bio Inc., Shiga, Japan). The column was washed with 10 volumes of PBS containing NaCl (300 mM) and imidazole (20 mM) to remove all contaminants. Subsequently, the protein was eluted from the column using PBS containing 250 mM imidazole. The eluted fractions were concentrated using a Nanosep Advance Centrifugal Device with 10K Omega (PALL Laboratory). Estimation of protein purity and quantity were achieved with SDS-PAGE and a Pierce BCA Protein Assay Kit (Thermo Fisher Scientific, Rockford, IL, USA), respectively.

### 2.6. FCoV-WV and SCoV2 RBD ELISAs with Overnight Serum Incubation

Corning ELISA plate wells were coated with 100µL of 100 μg/mL FCoV2-WV antigen or 100 μg/mL SCoV2 RBD antigen in sodium bicarbonate ELISA coating buffer at pH 9.5 (BioLegend, San Diego, CA, USA) and incubated overnight at RT. (Due to the limited amount and frequency of serum collection during COVID-19 between 25 March 2020 and 4 January 2021, ELISA with FCoV RBD antigen was not performed. Instead, more specific immunoblot analyses, using FCoV1 or FCoV2 RBD antigen, were conducted.) The next day, the plates were washed three times with phosphate-buffered saline tween (PBST). Non-specific binding sites were blocked with 100 µL per well of blocking solution (5% non-fat dry milk in sterile PBST- 0.5% Tween-20) for 1 h at 37 °C. After washing with PBST three times, 10 µL of cat serum was diluted in 0.990 mL of blocking solution (1:100) and incubated at RT overnight. After washing, horseradish peroxidase (HRP)-conjugated goat anti-cat IgG diluted 1:4000 (SouthernBioTech, Birmingham, AL, USA) in PBST was added and incubated at RT for 2 h. After washing, 100 µL of 3,3,5,5-tetramethylbenzidine High Sensitivity Substrate Solution (BioLegend) was added to the wells and incubated at RT for 15 min. The reaction was stopped by adding 100 µL of 1 N HCL in sterile water. The ELISA titer was measured at OD450 using BioTek’s Synergy HTX Multi-Mode Microplate Reader (BioTeK, Winooski, VT, USA).

### 2.7. Stringent FCoV-WV and SCoV2 RBD ELISAs

To ensure that the serum ELISA reactivity was specific to the FCoV2-WV or SCoV2-RBD antigen, sera were incubated individually at the same dilution for only 1 h instead of overnight. PBS was used instead of bicarbonate buffer for the coating of the antigen on the ELISA plates. Additionally, a BSA antigen control was included since veterinary vaccines used at the time in UGA queens often contained contaminating BSA from the cell cultures used during the manufacture of viral vaccines. This control was also important because the FCoV2-WV preparation contained about 5% BSA.

### 2.8. Gel and Immunoblot Analyses

The purified proteins were analyzed by SDS-PAGE and immunoblot. Briefly, the FCoV2-WV or RBD proteins (100 μg) were boiled individually at 95 °C for 5 min in a sample buffer. The protein(s) and prestained marker were loaded into individual wells, separated by 10% Tris-HCL gel with 30% or 40% acrylamide/bis, and stained with Coomassie blue solution for direct MW analysis or transferred to nitrocellulose membrane. Subsequently, the membrane was treated with Penta-His MAb (Qiagen, Germantown, MD, USA) followed by HRP-conjugated goat anti-mouse IgG (Invitrogen) and HRP substrate for degraded RBD protein/peptide distribution. In the serum reactivity studies, the FCoV2-WV or RBD proteins were evenly loaded into a 7 cm wide well of the stacking gel, with one 4-mm wide well at the end for the prestained marker, separated by 10% Tris-HCL gel with 30% or 40% acrylamide/bis, and transferred to nitrocellulose membrane. Each nitrocellulose blot was cut vertically into 26 3.2-mm width strips using a Novex Model NZ-1CIS strip cutter (Novel Experimental Technology, San Diego, CA, USA). Each immunoblot strip with 4.5 µg of RBD or FCoV2-WV was incubated individually with FCoV-infected cat serum or plasma at a dilution of 1:50 or 1:100 in a blocking buffer and incubated overnight at RT on a rocker. After three washes, the strips were then incubated with alkaline phosphatase (AP)-conjugated goat anti-cat IgG (1:1000) (SouthernBiotech) for 2 h at RT. Subsequently, after three washes, the reactive bands were visualized with freshly mixed AP substrate from an AP-Conjugate Substrate Kit (Bio-Rad).

### 2.9. FCoV2 NAb Assay against FCoV2

The Fc9 cells were used for both FCoV2 NAb studies and FCoV2/SCoV2 RBD blocking studies against live FCoV2 infection. We used a modification of the FCoV2 NAb assay as described previously [42]. Briefly, the diluted FCoV2 preparation (EMEM culture media with 5% heat-inactivated FBS) at 2 TCID_50_ was plated in a 96-well round-bottom microculture plate and then incubated with an equal volume of serially three-fold diluted cat serum. The plates were incubated at 37 °C in a 5% CO_2_ incubator for 45 min. The first row of the 12 wells was not used to prevent the drying effect. Thus, the second rows were the most concentrated serum dilutions of 1:6 with the virus preparation. The remaining serum dilutions in the wells were as follows: third rows with 1:18, fourth rows with 1:54, fifth rows with 1:162, sixth rows with 1:486, seventh rows with 1:1458, and eighth rows with 1:4374. Subsequently, 0.1 mL of the mixture of each well was transferred to the flat-bottom wells of an Fc9 cell monolayer with 95–97% confluency and incubated for 24 h at 37 °C and 5% CO_2_. The spent culture fluids were discarded. Each well was aliquoted with 0.1 mL of 0.25% sterile methyl cellulose in EMEM with 5% heat-inactivated FBS and incubated as before for 18 h or until the 35–50 virus plaques per well were observed. The FCoV2 plaques in the wells were inactivated and stained with 0.1 mL/well of 1% crystal violet in 100% methanol at RT for 10 min. Each well received an additional 0.15 mL of 1% crystal violet in 20% methanol, then was incubated at RT for 24 h and decanted. We then removed the stain with water.

### 2.10. RBD Blocking Assay against FCoV2

The RBD blocking assay against FCoV2 is a modification of the above FCoV2 NAb assay and differs from the NAb assay due to the following three features: (1) A set amount of RBD is used instead of cat serum. (2) FCoV2 dose was 4 TCID_50_ instead of 2 TCID50 to assure 95–100% cytopathic effect (CPE) on the Fc9 cells. (3) The flat-bottom plates with Fc9 cells at 98–100% confluency, instead of 95–97% confluency, were used when adding the virus mixture. The remaining procedure is identical to the NAb assay.

### 2.11. RBD Stimulation of PBMC from FCoV1-Infected Cats

The PBMC of transiently (4GC) and chronically (G-3) FCoV1-infected cats and two SPF cats (2FB, 4GA) were stimulated with 5 μg/mL of either FCoV1, FCoV2, or SCoV2 RBD in 0.1 mL RPMI 1640 media (Cat. #10040CM, Corning Inc., Corning, NY, USA). The PBMC was then supplemented with 10% heat-inactivated FBS and 50 μg/mL gentamycin per well in a round-bottom 96-well plate (Costar Cat. #3799, Corning Inc., NY, USA). The plate was incubated for 24 h at 37 °C and 5% CO_2_, and upon centrifugation, the spent culture fluids were discarded. The cell pellets were washed with PBS before total RNA was extracted with the Direct-zol RNA Micro-Prep method (Cat. R2063-A, Zymo Research Corp., Irvine, CA, USA), and the extracted RNA was reverse transcribed using LunaScript^®^ RT SuperMix Kit (Cat. NEB #E3010; New England Biolabs, Ipswich, MA, USA). The feline IFNγ mRNA were evaluated using primers (5′AATACCAGCTCCAGTAAACGG 3′ and 5′GCTTCCTCAGGTTAGATCTTGG 3′) and FAM-labeled probe (5′ FAM-CAGGTCCAGCGCAAAGCAATAAATGA-BHQ 3′) in a MIC qPCR machine (Bio Molecular Systems, Coomera, QLD, Australia). The feline glyceraldehyde-3-phosphate dehydrogenase (GAPDH) was used as the housekeeping gene expression using primers (5′ ATGTTCCAGTATGATTCCACCC 3′ and 5′ ACCAGCATCACCCCATTTG 3′) and FAM-labeled probe (5′ FAM-AAATTCCACGGCACAGTCAAGGC-BHQ 3′).

### 2.12. Data Availability

RBD sequences of FCoV1 RBD, FCoV2 RBD, CCoV1 RBD, CCoV2 RBD, human codon-optimized FCoV1 RBD, human codon-optimized FCoV2 RBD, and human codon-optimized SCoV2 UF2-RBD have been deposited in the NCBI database with accession numbers OP597272, OP597273, OP597274, OP597275, OP597277, OP597278, and OP597279, respectively. The complete sequences of the spike proteins used for our analyses and their NCBI accession numbers are shown in following Figure 2, and Appendix A.

## 3. Results

### 3.1. Initial Serological Studies in FCoV Naturally Infected Laboratory Cats

#### 3.1.1. FCoV Whole-Virus (WV) ELISA and SCoV2 ELISA with Overnight Serum Incubation

Our studies on FCoV were initiated when four laboratory queens donated by the UGA were mated with three SPF inbred toms from our laboratory at the University of Florida (UF) (Figure 1A). The four juvenile laboratory cats born from two UGA queens (UGAQ3 and UGAQ4) had FCoV Abs at 12 and 16 weeks of age based on FCoV2-WV ELISA. The UGA queens were all seropositive for FCoV2-WV (Figure 1B), and the sera from the toms, after mating, were weakly seropositive for FCoV2-WV but had no neutralizing Abs (NAbs) to live FCoV2 (Figure 1B). Since the FCoV2-WV has a large amount of NC in the preparation, sera from both FCoV1 and FCoV2 will react to NC in the FCoV2-WV ELISA [17]. Since these juvenile cats were to be used for a SCoV2 inoculation study at UGA, their serum was collected before shipment to UGA and tested by SCoV2 RBD ELISA. Three of the four juvenile cats had a serum that cross-reacted moderately with the SCoV2 RBD (Figure 1C).

#### 3.1.2. Stringent FCoV-WV ELISA and SCoV2 RBD ELISA

The sera from all four UGA queens and three toms were incubated at the same dilution for only one hour instead of overnight, and PBS was used instead of bicarbonate buffer for coating the antigen on the ELISA plate. In addition, ELISA with bovine serum albumin (BSA) antigen was included since the UGA queens were vaccinated three years ago with commercial vaccines. The veterinary vaccines were often contaminated with BSA. This BSA control was also important because the FCoV2-WV preparation contained about 5% BSA, whereas the RBDs were highly purified and devoid of BSA. Two UGA queens (UGAQ1 and UGAQ4) had high levels of serum reactivity with FCoV but without any reactivity with BSA and SCoV2 RBDs. The other two queens (UGAQ2 and UGAQ3) had high serum reactivities with FCoV, which were slightly higher than the serum reactivities with BSA (Figure 1D). The sera from UGAQ2 also had substantial cross-reactivities with both SCoV2 RBDs (UF and MassBiologics (MB) RBDs) (Appendix A), whereas the sera from UGAQ3 had a modest cross-reactivity with MB-RBD, below the threshold cross-reactivity with UF-RBD.

All UGA queens were group-housed together upon arrival on 25 October 2019. The earliest serum collected from the queens was on 19 December 2019 from UGAQ1 (Figure 1D). Hence, they were most likely infected with FCoV instead of SCoV2 and not from contact with a COVID-19-positive animal caretaker since the first case of COVID-19 in Florida was reported on 2 March 2020 [46]. The serum collected from 5HQT1 on 29 January 2020 (Figure 1E; post-2 mo) cross-reacted with SCoV2 RBD, supports FCoV infection based on the date of the first Florida cases. Furthermore, our cat facility has a strict personnel protective gowning (PPG) policy to protect our laboratory cats from external infections carried by the animal workers. Hence, the risk of having an SCoV2-asymptomatic animal worker transmitting SCoV2 to our laboratory SPF cats was minimal-to-none. However, our error was in trusting that UGA (a) purchased SPF cats from an accredited SPF vendor and (b) kept their animals free from FCoV, which was not the case.

As expected, the sera from all toms before mating had no reactivity with FCoV, both UF/MB-RBDs (Figure 1E) or BSA (Appendix A). However, post-mating sera from all toms had significant reactivity with SCoV2 RBDs but not with FCoV, compared to the corresponding pre-mating control results. The highest titers to SCoV2 RBDs were observed at the earliest timepoint of serum collection closest to the first exposure to the FCoV-positive queen. Subsequent sera showed declines that were still significantly different from the pre-serum titers (*p* < 0.05; pre- and post-exposure paired *t*-test). The decline suggested that the development of cross-reactive Abs to SCoV2 RBD could have occurred during active FCoV infection. A conflict was observed between the marginal reactivity with FCoV for the toms in Figure 1B (e.g., more non-specific binding with overnight serum incubation compared to one-hour serum incubation) and no reactivity with FCoV for the toms in Figure 1E. This may be due to the technical difference as detailed above. Overall, the stringent ELISA confirmed that the sera from all three toms cross-reacted strongly with SCoV2 RBD. Only one long-term FCoV-infected queen (UGAQ2) also had a high titer of cross-reactive Abs to SCoV2 RBD (Figure 1D).

### 3.2. The Lack of S1 and RBD aa Sequence Identity/Similarity between SCoV2 and FCoVs

#### 3.2.1. Sequence Analyses of SCoV2 and FCoV Structural Proteins

The aa sequence comparison between Wuhan SCoV2 versus FCoV1 and between SCoV2 versus FCoV2 demonstrated that SCoV2 structural proteins (spike, envelope, and membrane) were shorter than FCoV1 and FCoV2, with the exception of NC protein and S2 (Appendix A). The least aa identity and similarity were observed between SCoV2 and FCoV1/FCoV2, at the S1 glycoprotein, among the five structural proteins composing these viruses. The aa sequence identity and similarity between S1 glycoproteins of SCoV2 and FCoV1 had 13.0% and 40%, respectively, whereas, between SCoV2 and FCoV2, it was 16.3% and 43.4%. When comparing the cleavage sites separating S1/S2 among SCoV2, FCoV1, and FCoV2, the SCoV2 S1/S2 cleavage site appears closer to the FCoV1 S1/S2 cleavage site than that of FCoV2 [47,48]. In contrast, the S2 glycoprotein between SCoV2 and FCoV1/FCoV2 had the highest aa identity of 30.0% and 33.5%. This is the second highest similarity of 62.2% and 64.5%, respectively, among all structural proteins. Hence, S1 glycoprotein was the most distinctly different structural protein between SCoV2 and FCoV1/FCoV2 and among all five structural proteins.

When FCoV1 and FCoV2 spike glycoprotein sequences were compared, the S1 glycoprotein sequence had 29.5% and 31.6% identity and 62.9% and 64.5% similarity, which were much lower than the S2 sequence that had 60.6% and 68.7% identity and 84.0% and 89.9% similarity (Appendix A). The S1/S2 cleavage site for the FCoV2 is proposed to be at the S2 cleavage site (another cleavage site on S2 glycoprotein), next to the fusion peptide, because S1/S2 cleavage motif is absent in the location usually observed for FCoV1 and other coronaviruses [47,48,49]. Note that most coronaviruses, including SCoV2 and FCoV1, have the S2 cleavage site, but, for brevity, only S1/S2 cleavage site results were used in our analysis. The difference in the aa sequence between FCoV1 and FCoV2 at the envelope (Env), NC, and M ranged from 92.8–96.3% identity and 97.6–98.5% similarity, which demonstrated the high conservation between FCoV1 and FCoV2 for those structural proteins. The molecular weight (MW) of the structural proteins without glycosylation was predicted (Appendix A) to analyze the FCoV2-WV immunoblot described later. 

#### 3.2.2. Sequence Analyses of SCoV2 and FCoV RBDs

Next, the Wuhan RBDs of UF and MB were compared to the four known FCoV1 S1 sequences. Another set was compared to the four known FCoV2 S1 sequences, using Clustal O (1.2.1) multiple sequence alignment of the JustBio alignment server (https://justbio.com/ (accessed on 21 August 2020)) (Figure 2A,B). The goal of this alignment analysis was to identify the potential FCoV1 and FCoV2 RBD sites for use as vaccine immunogens. Our logic for using such a comparison is based on the fact that species-specific ACE2 is used by SCoV2 to infect cats, dogs, and humans. The Wuhan SCoV2 RBD sequence is distinctly different in aa sequences from the RBD sequences of FCoV1 (11.5% and 12.3% identity; 31.8% and 33.6% similarity) and FCoV2 (12.2% and 12.3% identity; 36.5% and 37.7% similarity) (Figure 2C). The first value is based on UF-RBD, and the second value is based on MB-RBD. As shown, the carboxyl-end of the UF-RBD sequence (residues 319–529) is shorter by 12 aa from the MB-RBD sequence (aa residues 319–541) (Figure 2A,B and Appendix A). The FCoV2 RBD had slightly more aa similarity with the Wuhan SCoV2 UF- and MB-RBDs than those between the FCoV1 RBD and SCoV2 RBDs (Figure 2C). Thus, the aa sequences of the two FCoV2 RBDs (36.5% and 37.7%), more than the two FCoV1 RBDs (31.8% and 33.6%), had slightly more similarity to SCoV2 UF- and MB-RBD sequences.

The full-length aa sequence comparison of S glycoproteins of the SCoV2 Wuhan strain and FCoV2 79-1146 strain displays the S1/S2 cleavage site for SCoV2 at a location different from the S1/S2 cleavage site for FCoV2 (Appendix A) [36,47,48]. In addition, the SCoV2 RBD sequence alignment pattern, with a single FCoV2 RBD sequence of the full-length S protein sequence, differs slightly from the one aligned with four FCoV2 RBD sequences (Figure 2B), even though both analyses used the JustBio alignment server. The full-length S sequences between SCoV2 and FCoV2 show that within the S1 sequence, the RBD has the least sequence similarity, followed by the N-terminal domain (NTD) and then C-terminal domain (CTD) (Appendix A). This pattern was the same whether the S1/S2 cleavage site for SCoV2 or FCoV2 was used. Additionally, the S2 sequence had the most sequence conservation between SCoV2 Wuhan and FCoV2 79-1146.

The full-length S sequences of the FCoV1 UCD-1 strain were compared to the S sequence of the SCoV2 Wuhan strain. They showed major changes in gap location on the SCoV2 sequence when compared to the single FCoV1 sequence (Appendix A) or to the four FCoV1 sequences (Figure 2A). The S1/S2 cleavage site for SCoV2 is only 46 aa plus five gaps away from the counterpart S1/S2 cleavage site for FCoV1 (Appendix A) [36,47]. Conversely, the FCoV1 S1/S2 cleavage site is 37 aa plus 14 gaps from the counterpart SCoV2 cleavage site [48]. Thus, the S1/S2 cleavage sites are closer to SCoV2 and FCoV1 than those between SCoV2 and FCoV2. The most aa sequence conservation is observed at the S2 sequence, followed by S1 CTD, S1 NTD, and S1 RBD, which is identical to those observed between SCoV2 and FCoV2.

The comparison of the proposed FCoV1 and FCoV2 RBDs shows three major gaps from the mid-to-carboxyl-end, which overlaps our predicted receptor binding motif (RBM) and has only 27.3% aa identity and 62.5% aa similarity (Figure 2D,E). The FCoV1 and FCoV2 at the proximity to our RBMs are quite different, which may explain why FCoV1 does not use fAPN as a host cell receptor. The aa mutation sites that lower or eliminate the neutralization activity of the reported six nMAbs are shown with three color-coded symbols for each nMAb below the FCoV2 RBD sequence (Figure 2D). The aa mutation sites are clustered at the mid-to-carboxyl-end of our proposed RBD, which indicate that FCoV2 neutralizing epitopes are at or in the proximity of our proposed RBM.

### 3.3. Immunoblot Analyses of Sera from Queens and Toms

The proposed FCoV2 RBD, with the most cross-reactive sectional regions overlapping with SCoV2 RBD, as highlighted in the magenta sequence (Figure 2B), were (1) produced in the same cell expression system with the same plasmid, (2) purified similarly as SCoV2 UF-RBD in PBS, and (3) used to develop the immunoblot strips. Two sera (UGAQ2 and UGAQ4) from the four queens and the serum from all three toms strongly cross-reacted with the SCoV2 UF-RBD (Figure 3A), but none of them reacted with the FCoV2 RBD (Figure 3B). All sera from the queens reacted strongly with FCoV2-WV immunoblot strips at the M (28–32 kDa), NC (43 kDa), degraded spike S (100–125 kDa), and entire spike S (190 kDa) (Figure 3C), with weaker bands at 20 kDa, 22 kDa, 40 kDa, and 55 kDa except for UGAQ1. The single serum available from UGAQ1 was collected when she was severely sick from uteritis and on antibiotics. Her serum only reacted to FCoV2-WV at 90 kDa and higher (Figure 3C) but not to SCoV2 (Figure 3A), FCoV2 (Figure 3B), and FCoV1 (Figure 3D) RBDs. The toms had reactivity to NC, M, and proteins at 10 kDa, 20 kDa, and 22 kDa but almost no reactivity to FCoV2-WV S proteins and degraded S glycoproteins above 90 kDa. The same sera from the toms did not react to FCoV2 RBD, except for one tom (HOGT3). HOGT3′s serum reacted weakly with FCoV2 RBD (Figure 3B) and without FCoV2 NAb titers (Figure 1B) but strongly with FCoV1 RBD (Figure 3D). These results demonstrate that the serum from three UGA queens (UGAQ2, UGAQ3, and UGAQ4) and all three toms reacted with FCoV1 RBD (Figure 3D). The nil-to-weak cross-reactivity to FCoV2 RBD and strong reactivity to FCoV1 RBD suggest that our toms and queens were infected with FCoV1, which is the most common serotype in the US. Their sera also had nil-to-minimum titers of FCoV2 NAbs (Figure 1B), which further supports our theory of FCoV1 infection of our queens and toms. One serum from a UGA cat (UGA1.4) (Figure 3B) with an FCoV2 NAb titer of 378 was our weak control serum that reacted to the FCoV2 RBD.

### 3.4. Characterization of Abs to SCoV2 RBD

Only sera from UGAQ2, UGAQ4 (10 months post-arrival), and all three toms had strong cross-reactivity to SCoV2 RBD (Figure 3A). Both 5HQT1 and HOGT3 were exposed to two FCoV-positive (FCoV+) queens over two breeding cycles (Figure 1A). HOJT2 was exposed to three different FCoV+ queens and retained the strongest SCoV2 RBD reactivity, even at 13 months post-first exposure to the first FCoV+ queen (Figure 3A). This observation suggested that this tom was actively producing SCoV2 RBD cross-reactive Abs, perhaps by sequential exposure to multiple FCoV+ queens. The queen UGAQ4 was exposed to three different FCoV+ toms during the three breeding cycles (Figure 1A) and had a moderate reactivity to SCoV2 RBD at 10 months post-first exposure (HOJT2), which was also 3 months post-second exposure (5HQT1) (Figure 3A). Nevertheless, UGAQ4 did not retain the cross-reactive Abs to SCoV2 RBD, even after the third exposure to the FCoV+ tom (HOGT3), 13 months post-first exposure (Figure 3A). This observation may suggest that the FCoV+ toms, upon exposure to FCoV+ queens, developed only low titers of FCoV infection based on the low reactivity of their FCoV2-WV Abs and nil-to-negligible serum reactivity to the bands above the predicted FCoV2 S2 band of 80–90 kDa (Figure 3C,E2). The rabbit polyclonal Abs to SCoV2 S protein and to S2 protein cross-reacted with FCoV2 S2, degraded S, and S glycoproteins on the FCoV2-WV immunoblot (Figure 3E1). This was not the case when rabbit polyclonal Abs to S1 protein was used. These results confirmed that the FCoV2 S, degraded S, and S2 MWs are detected by the sera from most UGAQ cats (Figure 3C,E2). Thus, the sera from the three toms, with strong cross-reactive Abs to SCoV2 RBD, negligible Abs to FCoV2 S, and degraded S, may suggest low FCoV1 titers in the toms. In addition, the possibility exists of cross-reactive SCoV2 RBD Abs lowering their level of FCoV1 infection.

### 3.5. Determining the FCoV2 Infection Blocking Activity of FCoV2 RBD

If the proposed FCoV2 RBD is indeed the RBD site for FCoV2, our FCoV2 RBD should be able to block the FCoV2 infection of the feline cell line (Fc9 cells). The duplicate wells, starting at 2.86 μg/mL, were serially diluted three-fold for each well, up to the titer of 0.106 μg/mL (fourth duplicate wells), and presented <50% cytopathic effect (CPE). All wells for virus control had 100% CPE including the PBS controls (Figure 4A). Thus, 1 TCID_50_ titer, or the titer with 50% CPE, is between 0.035–0.106 μg/mL. Another assay demonstrated 1 TCID_50_ titer of 50% blocking observed at about 0.100 μg/mL of FCoV2 RBD (Figure 4B). The SCoV2 RBD at 64.2 μg/mL blocked 100% of FCoV2 infection (first duplicate wells), whereas 42.8 μg/mL blocked 30% of CPE. Both FCoV2 and SCoV2 RBDs caused no cellular toxicity (Figure 4C). In addition, SCoV2 and FCoV2 RBD treatments, with the same doses, did not elicit cellular toxicity by decreasing the metabolism of the Fc9 cells, based on our preliminary MTT assay (data not shown). Overall, 0.1–1.0 μg/mL of FCoV2 RBD blocked 50–100% of FCoV2 infection, respectively, whereas 48–64 μg/mL of SCoV2 RBD cross-blocked 50–100% of FCoV2 infection, respectively, at doses without any cellular toxicity. Hence, the proposed FCoV2 RBD sequence includes the RBD site for FCoV2 infection.

### 3.6. Differential Reactivity to the FCoV1 and FCoV2 RBDs of the Plasma from the FCoV1 KU-2 and FCoV2 79-1146 Inoculated Laboratory Cats

All plasma from SPF cats inoculated with either FCoV1 KU-2 or FCoV2 79-1146 reacted to FCoV2-WV at NC, M, and S2 (Figure 4D). All plasma from cats inoculated with FCoV1 KU-2 strain strongly reacted with FCoV1 UCD-1 RBD (Figure 4E) but not with FCoV2 79-1146 RBD (Figure 4F). As expected, all plasma of cats inoculated with FCoV2 79-1146 strain strongly reacted with homologous FCoV2 79-1146 RBD (Figure 4F) but not with heterologous serotype FCoV1 UCD-1 RBD (Figure 4E). Since the majority of the FCoV1 transmitted cats developed cross-reactive Abs to SCoV2 RBD (Figure 3A), the fact that none of the plasma from FCoV1 KU-2 infected cats cross-reacted with SCoV2 RBD was unexpected (Figure 4G).

### 3.7. The Role of Feline Genetics and Group Housing in Inducing Cross-Reactive SCoV2 RBD Abs

Eight laboratory cats living together in group housing were all seropositive for FCoV infection by ELISA and immunoblot analyses (Appendix A). Since these cats share the litter pans and FCoV transmits by oral-fecal route [6,30,50], once the immunity to the first infection subsides, they can be susceptible to reinfection at different times based on the immune constitution of the cat. Cat G-2, with the highest FCoV Ab titer, had the strongest cross-reactive Abs to the SCoV2 UF-RBD immunoblot, followed by cats G-1, G-5, and G-7, with modest reactivity to SCoV2 RBD (Appendix A1). The remaining cats, G-4 and G-6, had negligible-to-no reactivity during the study, and cat G-3 showed no reactivity initially but 15 months later developed strong cross-reactivity to SCoV2 RBD. All eight cats were negative for FCoV2 RBD Abs throughout the study (Appendix A2). However, all sera were eventually positive at NC, M, and S2 protein bands by FCoV2-WV immunoblot within 15 months of group housing (Appendix A3). The cats appear to be all infected with FCoV1 based on a lack of reactivity to FCoV2 RBD (Appendix A2) and strong reactivity to FCoV1 RBD (Appendix A4). In our preliminary findings of group-housed pet cats, all three cats tested were Ab positive for FCoV by FCoV2-WV immunoblot (Appendix A1) and for FCoV1 by FCoV1 RBD immunoblot, while being negative for FCoV2 RBD (Appendix A2). Remarkably, one of them (KY at first and second bleeding 2.5 months later) was also positive for SCoV2 RBD.

### 3.8. Interferon-Gamma (IFNγ) mRNA Production of PBMC from FCoV1-Infected Cats upon Stimulation with SCoV2, FCoV1, or FCoV2 RBD

The PBMC from a transiently FCoV1-infected cat (4GC) and chronically FCoV1-infected cat (G-3) from the group-housed laboratory cats referenced above developed substantial levels of IFNγ mRNAs in response to SCoV2 RBD stimulation (Figure 4H). The levels were close to those stimulated by T-cell mitogen (concanavalin A, ConA). Chronically infected cat G-3 also responded with slightly lower, but substantial, IFNγ mRNA level in response to FCoV1 RBD stimulation, whereas transiently infected cat 4GC had no response to FCoV1 RBD. Both infected cats had no response to FCoV2 RBD stimulation. As expected, the PBMC from two uninfected SPF cats (2FB, 4GA) developed no IFNγ mRNA response to all three RBDs, while developing a robust IFNγ mRNA level to ConA stimulation.

## 4. Discussion

Our novel finding of cross-reactive Abs to SCoV2 RBD developing in cats during active FCoV infection was unexpected (Figure 1C–E), based on low aa sequence similarity at RBD of SCoV2 and FCoVs (Figure 2A–C). Hence, the original observation, using conventional SCoV2 RBD ELISA, was further analyzed and confirmed by more stringent SCoV2 RBD ELISA, immunoblot analysis, in vitro FCoV2 infection blocking assay with SCoV2 RBD, and, finally, cellular IFNγ immune responses to SCoV2 RBD. The peak sera from UGAQ2 and all three toms cross-reacted strongly with sensitive SCoV2 RBD ELISA and immunoblot but not with FCoV2 RBD (Figure 3A,B). Perhaps the most striking observation was the sera from the toms cross-reacting to SCoV2 RBD with nil-to-minimal reactivity to FCoV2 S glycoprotein on the FCoV2-WV immunoblot (Figure 3C). These results indicate that cross-reactive Abs to SCoV2 RBD appeared without major development of cross-reactive Abs to FCoV2 S glycoprotein. However, these results may be different if FCoV1 immunoblot was available for our use. None of our cats (queens, toms, and their kittens) developed major NAbs to FCoV2 based on FCoV2 79-1146 NAb assay (Figure 1B). This result further supports our theory that our cats were infected with FCoV1, which was later confirmed by their sera reacting strongly with FCoV1 RBD immunoblot strips (Figure 3D). In addition, the FCoV1 RBD immunoblot strips only reacted with plasma from FCoV1 KU-2 experimentally infected cats but not with plasma from FCoV2 79-1146 inoculated cats (Figure 4E). Conversely, FCoV2 RBD strips only reacted with sera from FCoV2-infected cats but not with plasma from FCoV1-infected cats (Figure 4F).

Unexpectedly, the plasma from FCoV1 KU-2 infected cats did not cross-react to SCoV2 RBD, which was not due to the use of plasma, since all plasma from FCoV2 79-1146 inoculated cats strongly cross-reacted with SCoV2 RBD (Figure 4G). This observation suggests that not all FCoV1-infected cats develop cross-reactive Abs to SCoV2 RBD, or the timing of the plasma collection was possibly too late (60 days post-inoculation). These factors, combined with a) low virus inoculation dose (1-mL/cat of 1 TCID_50_), b) oral/nasal inoculation route, and/or c) low S1/RBD immunogenicity of the infection, may have contributed to the lack of SCoV2 RBD cross-reactivity, similar to the lack/loss of cross-reactivity of UGAQ1 and UGAQ3 sera to SCoV2 RBD (Figure 3A).

The fact that our immunoblots are developed under reducing conditions supports our contention that the cross-reacting epitopes on SCoV2 RBD are either linear aa epitopes and/or glycosylated epitope(s). There are a sufficient number of sectional aa sequences similar between SCoV2 and FCoVs (Figure 2 and Appendix A), including the additional 12 aa sequence at the carboxyl-end of SCoV2 MB-RBD. Our preliminary FCoV1 RBD immunoblot results, with penta-His MAb (Figure 3G) combined with the studies using sera from FCoV1 transmitted or inoculated cats (Figure 3D and Figure 4E), demonstrate that Abs to the glycosylation component of the RBD are present in their sera. This possibility is important since NAbs to glycosylated epitopes have been identified for highly glycosylated HIV-1 envelope and SCoV2 spike glycoprotein [51,52].

Our SPF toms were highly inbred to recognize protective T-cell peptides of HIV-1 and FIV NC and reverse transcriptase, which contained many HLA-A2 and HLA-B27 peptides [53,54]. Hence, there was a remote possibility that our toms’ feline leukocyte antigen (FLA) genetics may have contributed to the unexpected cross-reacting Abs to SCoV2 RBD in all of our inbred toms. Furthermore, our inbred SPF toms were never vaccinated and lived in HEPA-filtered housing units, which prevented exposure to microbes found in the external environment. As a result, they are not highly immune-activated and do not produce high levels of antibodies to FCoV, compared to FCoV-infected pet cats. Consequently, we tested eight outbred laboratory cats living in group housing unrelated to our toms. All tested positive for FCoV infection, first by FCoV2-WV ELISA and later confirmed by FCoV2-WV immunoblot analysis (Appendix A). The majority of these cats had cross-reacting Abs to SCoV2 RBD, ranging from low to high titers (Appendix A1), which were maintained 15 months later. Moreover, SCoV2 RBD cross-reacting Abs developed in laboratory cats purchased from the same commercial cat vendor as the UGA queens but of different cat lineages. This supports the view that the laboratory cats sold by this commercial vendor have lineage(s) with FLA genetic makeup, which may sustain higher titers of cross-reactive Abs to SCoV2 RBD, such as cats UGAQ2 and G-2 (Figure 3A and Appendix A). Group housing appears to sustain the FCoV infection by reinfection without clinical episodes. Our preliminary results indicate that FCoV1-infected pet cats in the field possess cross-reactive Abs to SCoV2 RBD without manifesting clinical signs. These observations may also explain why only some SCoV2-infected pet cats display clinical signs while others remain asymptomatic. Such variation in clinical signs was also observed during SCoV2 infection of humans [55,56].

The lack of serum cross-reactivity with FCoV2 S glycoproteins by two of the three toms was unexpected since all four queens, including UGAQ2, reacted with FCoV2 S glycoprotein (Figure 3C). This is based on a rationale derived from the fact that both rabbit polyclonal Abs to S2 protein and to S protein cross-reacted with S, degraded S, and S2 of the FCoV2-WV (Figure 3E). S2 glycoprotein has high aa sequence identity and similarity. As a result, the likelihood of cross-reactivity with S2, rather than the S1 component of the S glycoprotein, is a strong possibility, as seen in the study by Zhao et al. (2019) [18]. Zhao et al. did not observe any serum from 137 FCoV-infected laboratory cats cross-reacting with SCoV1 S1 or other Betacoronaviruses S1 glycoproteins by ELISA. Their report shows serum from 15 FCoV-infected cats (10.9%) cross-reacting with HCCoV 229E S1 and the sera from two cats (1.5%) cross-reacting with HCCoV NL63 S1. Both HCCoV 229E and NL63 are human Alphacoronaviruses, and, as stated before, HCCoV 229E uses hAPN as the primary cell receptor [28,29], whereas HCCoV NL63 uses hACE2 as the primary cell receptor such as SCoV1 and SCoV2 [25,57]. None of the 137 sera from FCoV-infected cats cross-reacted with S1 of human SCoV1, MERS-CoV, HCCoV OC43, and HCCoV HKU1 [18]. HCCoV OC43 and HKU1 uses 9-O-acetyylated sialic acid and MERS-CoV2 uses dipeptidylpeptidase 4 (DPP4) as primary cell receptor [58,59]. This study, published in 2019, did not evaluate using the FCoV-WV immunoblot analysis or the cross-reactivity to SCoV2 RBD [19]. SCoV1 RBD has been reported to have 73% aa sequence identity to SCoV2 [35]. Our SCoV1 and SCoV2 S sequence analysis also confirmed their finding of 73% identity and further determined the aa sequence similarity of 90% at RBD and 91.9% at S glycoprotein (Appendix A). Our findings show that SCoV1 and SCoV2 S1 glycoproteins have 64.8% identity and 87.9% similarity (Appendix A). Therefore, our finding that FCoV-infected cats develop cross-reacting Abs to SCoV2 RBD was greatly unexpected when Zhao’s study showed no cross-reactivity to SCoV1 S1 glycoprotein.

Although in conflict with Zhao et al., which determined the lack of serum cross-reactivity of 137 FCoV-infected laboratory cats against SCoV1 S1 containing RBD, Hancock et al. (2022) reported 50% cross-reactivities of the pre-COVID-19 sera (2007–2012) from 109 domestic cats (70% client-owned and 30% feral) against only SCoV2 RBD by ELISA [60]. Our studies tested serum/plasma reactivities or cross-reactivities of laboratory cats infected with either FCoV1 or FCoV2 against FCoV1, FCoV2, and SCoV2 RBDs. We did so to clearly demonstrate that the sera from both serotypes of FCoV-infected cats cross-react to SCoV2 RBD, using the most specific immunoblot analyses. Hancock et al. do not define whether the SCoV2 cross-reactivity is caused by the conventional FCoVs or their FCoV serotype(s) [60]. Our study further demonstrates that the duration of serum cross-reactivities to SCoV2 RBD is short in months (Figure 1D,E on UF-RBD; Figure 3A), even if the infected laboratory cats are housed together (Appendix A1) and for the single-household, infected pet cats (Appendix A2). Therefore, their 50% cross-reactivities of the pre-COVID-19 cat sera appear to be too high, and such a percentage may be attributed to non-specific reactivity to BSA. Unfortunately, Hancock et al. did not test the sera for BSA reactivity by ELISA when the client-owned cats are most likely vaccinated with commercial feline vaccines containing BSA, as shown in our study (Figure 1D). Hancock et al. only tested SCoV2 RBD purified by Ni-NTA resin [60], whereas our sensitive immunoblot studies tested SCoV2 RBD purified by TALON (cobalt) resin. TALON resin has more specificity to His aligned consecutively, as in His-tagged protein, than Ni-NTA resin [61,62]. The non-specific binding of Ni-NTA to BSA can occur more readily because the 5% fetal bovine serum of the culture media used to express their RBD has an ample supply of BSA [60]. Our work used serum-free Expi293 Expression Medium without any supplement [63]. BSA has 16 scattered His residues which can dimerize in buffer at low temperatures, especially in PBS [64,65], causing His residues in proximity to each other to increase the incident of non-specific binding to Ni-NTA [61,62]. Hence, their purified RBD most likely had BSA contamination, based on their control gels with an undefined weak 67–70 dKa band [60]. Hancock et al. should have used immunoblot analysis, which can distinguish the BSA band (67 kDa) from their RBD band (32–35 dKa) and confirm their SCoV2 RBD cross-reactivities in all 109 sera, or at least with the 50 cross-reactive sera. Nevertheless, our immunoblot work supports their original ELISA findings that at least a few or some of them are due to FCoV. Furthermore, their work supports our findings that more than known single-household, client-owned pet cats are seropositive for SCoV2 RBD in the US.

The use of SCoV2 RBD to block the in vitro infection of SCoV2 has already been performed and demonstrated by others by blocking the SCoV2 infection of Vero CCL-81 cells with SCoV2 RBD-His tag at IC50 of 21 μg/mL [66]. The novelty of our study is that SCoV2 UF-RBD at a high concentration was able to cross-block or cross-protect against FCoV2 infection of feline Fc9 cell line, at an RBD dose of 48–68 μg/mL without any cellular toxicity (Figure 4A). This is slightly more than twice the dose used by Shin et al. to block SCoV2 infection of Vero cells with SCoV2 RBD-His tag [66]. The ability of SCoV2 RBD to cross-protect against in vitro FCoV2 infection, although weaker than that of FCoV2 RBD, further supports our finding that SCoV2 RBD and FCoV2 RBD may be structurally and antigenically similar. We were unable to perform a similar blocking study against FCoV1 infection with SCoV2 and FCoV1 RBDs due to the lack of an FCoV1-susceptible cell line available [6,29]. Instead, the PBMC from transiently and chronically FCoV1-infected cats was used to evaluate whether SCoV2 and FCoV1 RBDs were recognized by the T cells in the PBMC from FCoV1-infected cats. Remarkably, the PBMC of both infected cats recognized SCoV2 RBD by producing IFNγ mRNA in response to its stimulation. However, the FCoV1 RBD stimulation was recognized only by the chronically infected cat. Although a larger number of animals is needed to confirm the results, this observation may suggest that the epitopes in the SCoV2 RBD are similar to those of FCoV1 RBD, while others are possibly more inflammatory than those of FCoV1 RBD. Since IFNγ is mainly produced by CD4^+^ and CD8^+^ T cells in the PBMC preparation [67], our observation suggests that the T cells recognize these RBDs as having similar T-cell epitopes.

Both FCoV1 KU-2 and FCoV1 UCD-1 are FIPV1s isolated from symptomatic cats and known to replicate in Fcwf-4 cells [68,69,70]. The culture-adapted FCoV1 UCD-1 has been shown to use heparin sulfate as the primary receptor to replicate in Fcwf-4 cells [70]. Our FCoV1 RBD is derived from the FCoV1 UCD-1 spike sequence, submitted to NCBI GenBank by the Japanese research team. This sequence was based on the FCoV1 UCD-1 grown in Fcwf-4 cells provided by the University of California, Davis (UCD). Hence, our finding that the plasma from FCoV1 KU-2-infected laboratory cats is reacting to our FCoV1 RBD was expected since KU-2 most likely uses the same primary receptor to infect Fcwf-4 cells. Our current study reveals that the sera from naturally FCoV1-infected pet cats can also react to our UCD-1-based FCoV1 RBD. This finding suggests that our FCoV1 RBD sequence may interact with both the heparin sulfate receptor and the unknown natural host cell receptor. It would be interesting to perform the blocking study on FCoV1 KU-2 or on culture-adapted FCoV1 UCD-1. More importantly, we need to test if culture-adapted UCD-1-infected SPF cats develop cross-reactive antibodies to SCoV2 RBD at the early stage of infection. Unfortunately, our laboratories at UF do not have any FCoV1/FIPV1, including KU-2 or UCD-1, to use in an RBD blocking study or to develop the FCoV1-WV immunoblots. Furthermore, ATCC does not sell FCoV1 or FIPV1.

An interesting observation is that Shin et al. used SCoV2 Wuhan RBD, identical to MB-RBD, with the extra 12 aa that had a higher aa sequence similarity of SCoV2 RBD, with both FCoV1 and FCoV2 RBDs. Shin et al. determined that the RBD-Fc tag was better at in vitro inhibition of SCoV2 infection than the RBD-His tag [66]. FCoV2 RBD had extended amino end and blocked infection more efficiently. Hence, we reasoned that a slightly larger SCoV2 RBD might block the in vitro infection against SCoV2 and FCoV2 better than a CoV-non-specific tag as long as it retains the native conformation. Such RBDs, in theory, must increase anti-SCoV2 cytotoxic T lymphocyte (CTL) epitopes without increasing T-helper epitopes for inflammatory responses. Remarkably, our preliminary result shows SCoV2 UF2-RBD (gp45) with strong reactivity to sera from COVID-19-vaccinated humans may achieve these conditions (Appendix A; Appendix A). Pfizer SCoV2 RBD mRNA vaccine was not as effective as their S mRNA vaccine, perhaps due to the shorter RBD sequence and the T4 foldon used in place of the SCoV2 S2 glycoprotein [71,72]. The Pfizer team reports that the immunogenicity against SCoV2 is similar, but the reactogenicities and adverse effects are slightly higher with the RBD mRNA vaccine than with the S mRNA vaccine [72]. As to the importance of S2 glycoprotein, weak linear NAb epitopes have been identified by the sera from COVID-19 patients on the amino end of the S2 glycoprotein, overlapping the fusion peptide (FP) next to the S2 cleavage site (Appendix A) [73,74]. The FP sequences (11–29 aa) are highly conserved among animal and human α/β-CoVs, including FCoV1/2 and CCoV1/2. Additionally, five MAbs, with broad NAb activity against human β-CoVs, were isolated from SCoV2-infected subjects [75]. They reacted to the conserved S2 stem helix (14–24 aa), potentially to sterically block the membrane fusion required for viral entry. Based on our analyses, FCoV1 and FCoV2 have 78.6% and 71.4% aa similarity to the 14 aa sequence of the SCoV2 stem helix core (Appendix A). A section of SCoV2 Delta HR1-HR2 linked to the Delta RBD as RBD-HR successfully assembled into a trimer configuration in cell culture [76]. Subsequently, their RBD-HR/trimer protein mixed in MF59 adjuvant was tested as Delta RBD-HR/trimer vaccine in multiple animal models against Delta and Omicron variants and showed protective efficacy. However, the use of human cell-based protein vaccines in animal models can cause artificial xenogeneic protection which has been reported [77,78,79]. Thus, an mRNA or DNA vaccine approach will remove such problems. As more serious adverse effects are reported for Pfizer and Moderna S mRNA vaccines [80], the reactogenicity/inflammatory epitopes must be removed from both SCoV2 S1 and S2 glycoprotein expressed upon vaccination. Hence, the minimalistic approach, such as RBD-HR, may induce fewer adverse effects since the S2 platform has changed from T4 foldon to HR1-HR2, which may alter the structural conformation of the RBD expressed.

The ability of FCoV2 and SCoV2 RBDs to block in vitro FCoV2 infection and also induce pan-CoV-specific T-cell responses suggests that these RBDs may be important for developing an effective pan-coronavirus vaccine for pet animals such as cats, dogs, and hamsters. SCoV2-infected hamsters from Europe imported to pet shops in Hong Kong have been reported as a source of two separate hamster-to-human transmissions and subsequent human-to-human transmission of the SCoV2 Delta variant, with the sequence found predominantly in Europe [11]. Inoculation of laboratory Syrian golden hamsters resulted in infection of the hamsters, with a major loss in weight, lung infection, and respiratory disease [81,82]. Although not SCoV2, Lednicky et al. reported porcine Deltacoronavirus infection in three children in Haiti with clinical symptoms of fever and two children with coughing and abdominal pain [83]. Lednicky et al. also discovered a US citizen visiting Haiti who developed fever and malaise [84]. This individual was diagnosed with a coronavirus infection resembling a recombinant of predominantly CCoV with sequences similar to CCoV-HuPn-2018. CCoV-HuPn-2018 appears to be a recombinant of FCoV2 and CCoV isolated from a patient with pneumonia in East Malaysia [7]. Based on these findings, a pan-CoV vaccine that prevents active FCoV and CCoV infections of cats and dogs will indirectly prevent infection of humans, which is another vital role of such a vaccine. Furthermore, a recent publication demonstrates the cat-to-human transmission of the SCoV2 Delta variant [85]. Thus, the development of pan-CoV vaccines that prevent infection in cats and humans is urgently needed.

Besides the profound acute severe respiratory syndrome, SCoV2 infection in humans causes gastrointestinal (GI) manifestations (diarrhea, vomiting, nausea, and abdominal pain), including longer fecal shedding than those detected in the nasopharyngeal samples [10,86,87]. Both FCoV and CCoV also cause GI tract disease in their respective animal hosts and clinically affect kittens and puppies more than adults, with the exception of FIPV disease [7,24,87]. FCoV2 RBD sequence alignment comparison of CCoV serotype 2 (CCoV2) and FCoV2 RBDs shows 95.6% aa sequence similarity and 87.7% aa sequence identity (Appendix A). The high sequence similarity between FCoV2 and CCoV2 RBDs may explain why CCoV2 can infect fAPN-expressing feline cells [27]. Although FCoV1 and CCoV1 RBDs possess only an aa sequence identity of 55.5%, their sequence similarity of 80.8% is remarkably high (Appendix A), suggesting that they have a common lineage with evolutionary changes and perhaps also by sectional recombination [9,29,88].

The first blood collection from our tom was on 29 January 2020, which was only one month before the first human case of COVID-19 in Florida [46]. Consequently, our laboratories at UF pondered on how to distinguish SCoV2 infection from FCoV infection in cats when only the sera from the day of euthanasia were available in reasonable amounts. We reasoned that if our toms were instead infected with SCoV2, then the SCoV2 RBD Abs should last longer with stronger reactivity than the FCoV1 RBD Abs and vice versa. A titration analysis was performed on the last serum from both toms and queens. Our toms’ sera on the day of euthanasia retained antibodies to FCoV1 RBD but not to SCoV2 RBD (Appendix A). As expected from the toms’ results, sera from UGAQ2 and UGAQ4 had high to no Abs to SCoV2 RBD, respectively, at 1:50 dilution, whereas both sera had high Ab titers to FCoV1 RBD, even at 1:1000 dilution (Appendix A). Furthermore, as expected from the negative results of UGAQ3 in Figure 3A, the serum from UGAQ3 had no reactivity to FCoV1 RBD and also no cross-reactivity to SCoV2 RBD. Hence, the SCoV2 Abs of both the queens and toms were indeed cross-reactive Abs, which were caused by the FCoV1 infection and not by the SCoV2 infection. This titration analysis further confirmed that, in FCoV1-infected cats, the cross-reactive Abs to SCoV2 RBD are lost first, before the Abs to FCoV1 RBD. A similar clinical case was received by our program, seen in an 8-month-old, female pet cat with severe lung pathology. The cat’s affected lung tissue was negative for FCoV RT-PCR, and her serum was negative for FCoV Abs, as confirmed by ELISA performed by other diagnostic laboratories. Since lung pathology is common for both SCoV2 and FIPV infections, our program agreed to perform the FCoV2-WV/RBD immunoblot analyses on the patient (AP) and later on her brother (HP) and another older cat (MP) living in the same household. All three cats were infected with FCoV1 but not with FCoV2 (Appendix A1,C2). The patient and her brother, cat HP, also had high Ab reactivity to SCoV2 RBD but not cat MA, suggesting that they were most likely infected with FCoV1 and not with SCoV2. Cat MP, who is without cross-reactive Ab band to SCoV2 RBD, is clearing FCoV1 infection more rapidly than cat HP. The patient’s serum also reacted strongly to the BSA band at 67–70 kDa of the FCoV2-WV immunoblot strip (Appendix A). Her serum was also tested for FIV Abs by FIV-WV immunoblot analysis and was negative for FIV Abs but positive for anti-BSA Abs (Appendix A). The false negative by FCoV ELISA for patient AP was probably due to the anti-BSA Abs requiring correction of the value for FCoV Ab titers. This second pet cat household study confirms that FCoV1 infection does cause cross-reactive Abs to SCoV2 RBD to develop, most likely during the peak of FCoV1 infection and clears before FCoV1 RBD Abs (Appendix A).

Since FCoV infection has existed in domestic cats for over three decades [4,5], the FCoV RBDs must have endured the evolutionary pressure and are unlikely to undergo further mutations at RBD to retain their infectivity. Current studies did not evaluate the possibility of FCoV1/FCoV2 coinfection, but such scenarios do occur in nature [89,90]. Hence, combining both FCoV1 and FCoV2 RBDs with the SCoV2 RBD will be essential for developing a pan-CoV vaccine for cats. Based on these findings, we propose that a pan-CoV vaccine against SCoV2 infection in cats, dogs, and hamsters can be developed by combining FCoV1 RBD (gp52) and FCoV2 RBD (gp59) together with a larger SCoV2 RBD (gp45) for stability, potentially without the addition of CCoV1/CCoV2 RBDs. However, it may be necessary to add conserved pan-CoV CTL epitopes from highly conserved SCoV2/FCoV protein(s) (e.g., RNA-dependent RNA polymerase with minimal mutations among SCoV2 variants) to the triple-RBDs, each in a configuration of RBD-HR [76] as an mRNA or DNA vaccine, to induce sterilizing immunity (i.e., immunity against infection) in the vaccinated animals. Such pan-CoV vaccines, tested in laboratory cats, may provide insights on how to develop sterilizing immunity against SCoV2 in humans while also preventing future zoonotic variant infections and benefiting companion animals.

## 5. Conclusions

This report describes the presence of cross-reactive antibodies to SCoV2 RBD in the sera of laboratory cats infected with well-characterized FCoV1 and FCoV2 strains from pre-COVID-19 pandemic years. Thus, cross-reactivity was not caused by rare CoV variants. Sequence analyses of the SCoV2, FCoV1, and FCoV2 RBDs demonstrated minimum aa sequence identity and similarity among them. Current studies confirmed our original ELISA results of SCoV2 cross-reactivity. This confirmation was achieved by using more sensitive immunoblot analyses with the sera from 25 FCoV-infected laboratory cats against all three RBDs and FCoV2-WV, SCoV2 RBD blocking of the in vitro FCoV2 infection, and the positive SCoV2 RBD stimulation, most likely, of the T cells in the PBMC from FCoV1-infected laboratory cats. Other original findings are as follows: (1) The current study utilized a unique approach by using the SCoV2 RBD sequence to identify the FCoV1 and FCoV2 RBDs successfully. (2) FCoV2 RBD blocked the in vitro FCoV2 infection by at least 64-fold more than SCoV2 RBD, demonstrating that the FCoV2 RBD produced indeed includes the RBD site for FCoV2 infection and is potentially useful as vaccine immunogen against FCoV2. (3) The sera/plasmas from FCoV1-infected cats reacted only to the FCoV1 RBD, whereas those from the FCoV2-infected cats reacted only to the FCoV2 RBDs. This suggests that both FCoV serotype RBDs are required as vaccine immunogens against FCoVs. (4) The discussion provides the strategy on how to produce a pan-CoV vaccine with sterilizing immunity to SCoV2 and FCoVs. (5) The discussion also provides the current findings on a SCoV2 RBD vaccine trial and a natural HR1-HR2 platform for RBD, available for the design of a pan-CoV vaccine for humans, with less adverse effects than FDA-approved COVID-19 vaccines. Therefore, current findings will lend insight into the development of a pan-CoV vaccine for animals and humans.

## Figures and Tables

**Figure 1 viruses-15-00914-f001:**
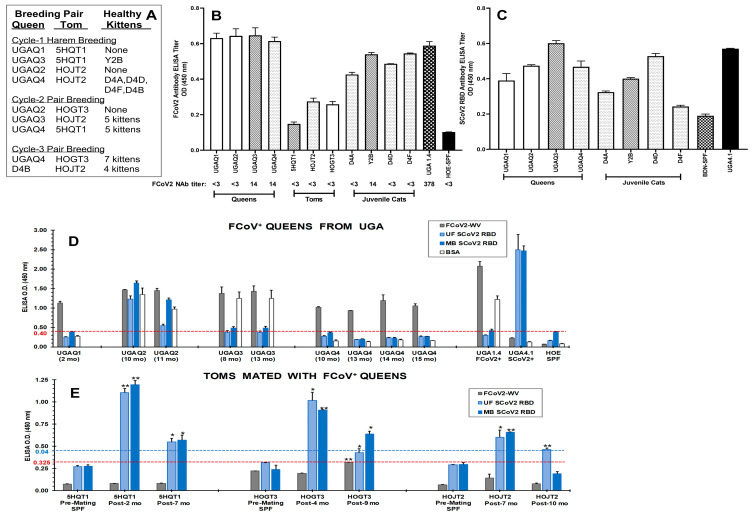
Serum Ab reactivity of the FCoV-positive cats to FCoV2-WV and SCoV2 RBD antigens by ELISA. Four queens (UGAQ1, UGAQ2, UGAQ3, and UGAQ4) were mated at UF with three SPF toms (5HQT1, HOJT2, and HOGT3). In the first harem breeding cycle (**A**), only two of the four queens (UGAQ3 and UGAQ4) successfully mated with two different toms (5HQT1 and HOJT2) and gave birth to kittens. Queen UGAQ3 mated with 5HQT1 and gave birth to Y2B (**A**), and their bars are represented with slashes (**B**,**C**). Queen UGAQ4 mated with HOJT2 and gave birth to D4A, D4D, D4F, and D4B (**A**). The bars of D4A, D4D, and D4F are represented with small dots (**B**,**C**). At the time of first blood collection, these kittens were juvenile cats at 16 weeks old for D4A, D4D, and D4F and 12 weeks old for Y2B (**A**). The open bars are the two queens UGAQ1 and UGAQ2 (**B**,**C**) that were also harem mated with toms 5HQT1 and HOJT2, respectively. They did not give birth to kittens in the first mating or in the pair mating for UGAQ2 (**A**). The serum from FCoV2-seropositive cat UGA1.4 (**B**) and SCoV2-inoculated cat UGA4.1 (**C**) was used as positive control serum. The serum from SPF cats HOE (**B**) and BDN (**C**) were used as negative control serum. The FCoV2 NAb titers shown below the cat identification code (**B**) were those measured at first arrival for queens and post-first mating for toms. FCoV2 NAb assays were performed thrice for repeatability. The sera from the later timepoints described in Figure 1D,E were tested twice and had no detectable NAbs (data not shown). Additionally, the serum Ab cross-reactivity of four FCoV^+^ queens (**D**) at post-UF arrival in months (mo) and three toms (**E**) at pre-mating and post-first mating with the FCoV^+^ queens was tested by stringent ELISA. Their sera were tested for reactivity to FCoV2-WV (grey bar), SCoV2 UF-RBD (light blue bar), SCoV2 MB-RBD (blue bar), and BSA (white bar). The positive threshold value for Figure 1D is 0.400 O.D., based on the maximum, non-specific binding of the serum from the SPF cat HOE to the antigen, MB SCoV2 RBD. Since the pre-mating serum was collected from the toms when they were still SPF, we used the pre-mating serum from an SPF tom with the highest non-specific reactivity to antigen. The positive threshold value (dotted red line) in Figure 1E is set at 0.325 O.D., which is based on the highest reactivity of the pre-mating serum from the SPF tom HOGT3 to the antigen FCoV2-WV. The positive threshold value (dotted blue line), which is higher than the pre-serum threshold, is based on the highest non-specific binding of the serum from the SPF cat HOE, which is the same as Figure 1D. Significant difference between the pre-mating and post-mating sera of the respective bars are shown as *p* < 0.05 (*) or *p* < 0.005 (**) (**E**). The serum from SPF cats HOE and BDN was used as a negative control (**B**,**D**,**E**) and (**C**), respectively. The serum from cat UGA1.4 and UGA4.1 were used as an FCoV2 positive control (**B**,**D**,**E**) and SCoV2 positive control (**C**–**E**), respectively. The same sera from the toms tested negative for anti-BSA Ab titers before this set of ELISA assays (Appendix A).

**Figure 2 viruses-15-00914-f002:**
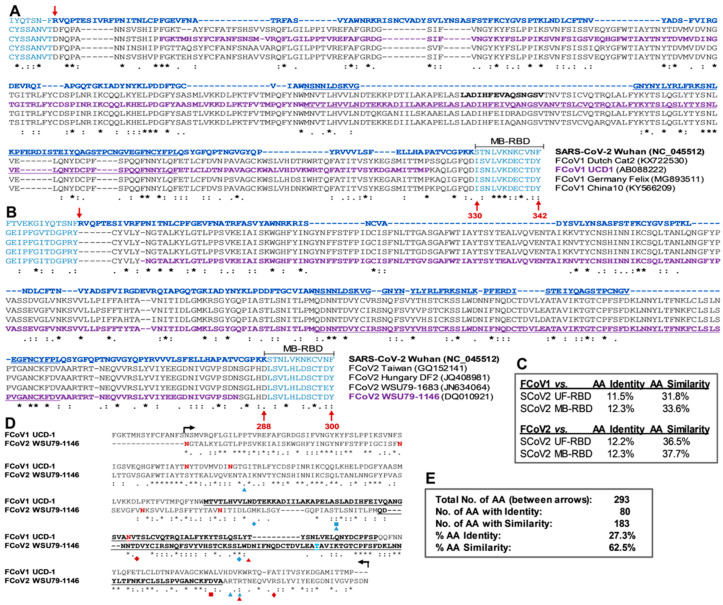
Amino acid (aa) sequence alignment of SCoV2 Wuhan RBD along with proposed FCoV1 RBD (**A**) and with proposed FCoV2 RBD (**B**) using multi-alignment analysis with four FCoV strains. In order to prevent one strain anomaly, four strains of FCoV1 (**A**) were aligned with SCoV2 Wuhan RBD by Clustal Omega 1.2.1 of JustBio Server (https://justbio.com/). The SCoV2 Wuhan RBD sequence is shown at the top in blue, and the SCoV2 receptor binding motif (RBM) is underlined. Magenta-colored RBD sequence belongs to the proposed FCoV1 UCD-1 RBD sequence (**A**, second FCoV1 sequence) and the proposed FCoV2 79-1146 RBD sequence (**B**, fourth FCoV2 sequence). The blue-labeled residues below the MB-RBD bracket represent MassBiologic’s SCoV2 RBD with the extra 12 residues. The identical and similar aa residues are shown with an asterisk (*) for complete identity, strong similarity with a colon (:), and modest similarity with a single dot (.). The gaps are shown with a dash (-). The aa sequence identity and similarity for UF-RBD (without carboxyl-end 12 aa residues) and MB-RBD sequences are summarized (**C**), starting with arginine (R) on the amino end of SCoV2, pointed downward with a red arrow, and ending with an upward red arrow, with the number of aa for UF-RBD or MB-RBD with gaps. Next, the proposed FCoV1 UCD-1 RBD sequence (aa residues 406–684) and the proposed FCoV2 79-1146 RBD sequence (aa residues 408–675) were aligned similarly using JustBio Server (**D**). The bolded, underlined sections represent potential RBM regions based on counterpart SCoV2 RBM regions in Appendix A for FCoV2 79-1146 and Appendix A for FCoV1 UCD-1. The FCoV1 UCD-1 RBD has three N-glycosylation sites with a high prediction, as shown with a bolded red N and no O-glycosylation, based on the NetNGlyc 1.0 Server and NetOGlyc 4.1 server, respectively (network address in Appendix A legend). FCoV2 79-1146 RBD has four predicted N-glycosylation sites (N), shown in bold red, and one O-glycosylation site in bold bright blue (T). The aa sequence identity and similarity between the two RBD sequences are summarized (**E**), starting with asparagine (N), with a start arrow and ending with an end arrow. The six color-coded symbols below the FCoV2 79-1146 sequence represent six nMAbs identified by Kida et al. in blue [41] and Corapi et al. in red [42], and these nMAbs were produced using FCoV2 79-1146. The three nMAbs of Kida et al. do not cross-neutralize FCoV1 virus (T. Hohdatsu, coauthor of this article and reference [41]).

**Figure 3 viruses-15-00914-f003:**
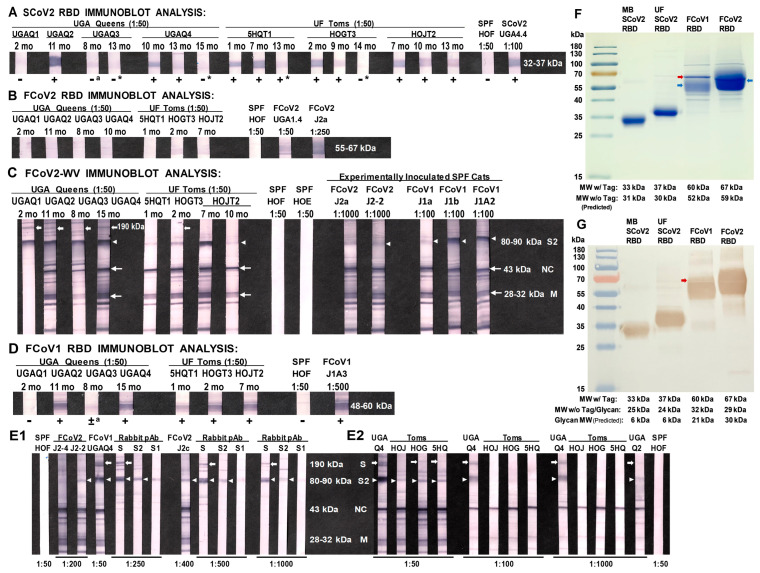
Immunoblot analyses of the sera from FCoV^+^ queens and SCoV2 RBD-positive toms followed by MW analyses of the RBDs. Immunoblots were developed for the SCoV2 UF-RBD (**A**), the proposed FCoV2 RBD (**B**), the cross-reactive FCoV2-WV (**C**), and the proposed FCoV1 RBD (**D**). The results for SCoV2 UF-RBD (**A**) and FCoV1 RBD (**D**) are shown below the immunoblot strip, with the summary result of either a positive (+) or negative (−). Those with an asterisk (*) shown were supported by the result in Appendix A, and those with a subscript “a” shown were supported by the result performed at 1:25 dilution (data not shown). Similar to Figure 1D,E, the time of serum collection from four FCoV^+^ queens is shown as time post arrival to UF, and those from the toms are shown as time post-first exposure to an FCoV^+^ queen. The MW size range for the RBD bands SCoV2, FCoV2, and FCoV1 are shown next to the control bands. The bands for FCoV2 structural proteins in FCoV2-WV immunoblot (**C**) are 43 kDa for NC (long arrow), 28–32 kDa for M (long arrow), and 80–90 kDa for spike-2 glycoprotein (S2) (arrow head). The 190 kDa band (short arrow) in the FCoV2-WV immunoblot strips is the S glycoprotein, confirmed by a cross-reactivity analysis using rabbit polyclonal antibodies (pAb) to SCoV2 S2 (Sigma ABF1063) and SCoV2 S (Sigma ABF1066), which cross-reacted with FCoV2 S2 and S, respectively (**E1**). The rabbit pAb to SCoV2 S1 (Sigma ABF1065) did not cross-react with FCoV2 S1 on immunoblot. Consequently, it served as rabbit control pAb to detect non-specific trapping of the rabbit pAb to the FCoV2 S2 band, which had the highest antigen load on the immunoblot. The same analysis was performed at three serum dilutions for UGAQ4 and three toms (HOJT2, HOGT3, and 5HQT1), with UGAQ2 only at 1:1000 using the immunoblot of the same batch (**E2**). The sera collected on the day of euthanasia were used for UGAQ4 and the three toms (**E2**). All immunoblot photographs were adjusted for consistency to 10% brightness and 5% contrast. The actual MW of the RBDs used in the immunoblot analyses was determined by Coomassie blue staining of RBD gel (**F**) and Penta-His MAb treated RBD immunoblot (**G**). The SCoV2 UF-RBD, FCoV1 RBD, and FCoV2 RBD used the same plasmid with cleavage site peptide and two tags (70 aa with MW of 7454 Da) attached on the carboxyl-end (Appendix A). The SCoV2 MB-RBD had c-Myc peptide tag and 6× His tag (16 aa with MW of 1202 Da) besides the additional 12 aa extension of the RBD (Appendix A). The predicted MW without attachment (w/o tag) (**F**), predicted MW without tag and glycan (**G**), and the predicted glycan MW (**G**) are shown. The blue arrows represent the location of the peptide band within the major glycosylation band (**F**). The red arrow points to the sharp thin glycosylated band (**F**), which is an extremely thin peptide line indicated by a red arrow in the immunoblot (**G**). These results, highlighted with the red arrows and the thin glycosylated band seen immediately above the major glycosylated peptide band in panel D, indicate that FCoV1-infected cats also develop antibodies to the glycosylation component of the RBD.

**Figure 4 viruses-15-00914-f004:**
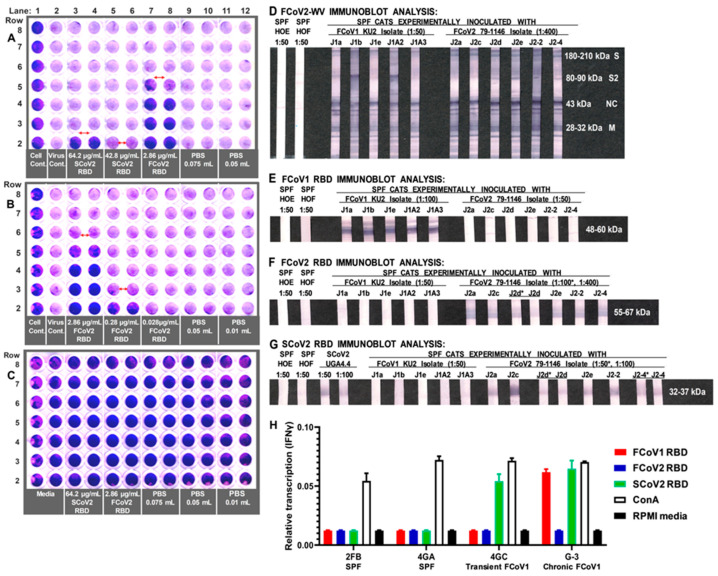
The in vitro FCoV2 infection-blocking activity of the proposed FCoV2 RBD and the SCoV2 RBD followed by the immunoblot analyses of the sera from FCoV-inoculated cats. In Plate A (**A**), the lanes consisted of media control without virus (rows 2–8, lane 1), virus control (rows 2–8, lane 2), SCoV2 RBD starting at 64.2 ug/mL (row 2, lanes 3–4), SCoV2 RBD starting at 42.8 ug/mL (row 2, lanes 5–6), FCoV2 RBD starting at 2.86 ug/mL (row 2, lanes 7–8), 0.075 mL of PBS in the first well (row 2, lanes 9–10), and 0.05 mL PBS in the first well (row 2, lanes 11–12). Each well in lanes 3–12, starting from row 3, are serial three-fold dilutions of the RBD or PBS. The RBD stocks were suspended in PBS. The PBS lanes represent the largest volume of the SCoV2 RBD (0.075 mL) and FCoV2 RBD (0.050 mL) used in the wells in row 2. Plate B (**B**) used the same plating scheme but had duplicate wells of FCoV2 RBD in ten-fold dilution in row 2, as described at the bottom. The 0.05 mL PBS control represents the largest volume of FCoV2 RBD used. The red horizontal double arrow shows where the blocking effect of the RBD stops when directly observed at the plate in plates A and B. The control Plate C (**C**) shows the highest amounts used for SCoV2 RBD and FCoV2 RBD for Plate A, tested without virus, to determine the potential cell toxicity caused by the RBDs. After scanning the plates at 100% brightness and 70% contrast, the plates were additionally brightened by 25% for plates A and C and 30% for Plate B, with all plates at an additional 3% contrast. Next, the plasma from five FCoV1 KU-2-inoculated laboratory cats and six FCoV2 79-1146-inoculated laboratory cats were tested for their immunoblot reactivity to FCoV2-WV (**D**), FCoV1 RBD (**E**), FCoV2 RBD (**F**), and SCoV2 RBD (**G**). The serum from two SPF cats, HOE and HOF, served as the negative controls (D-G). Additional positive control serum from SCoV2-inoculated cat UGA4.4, at 1:50 and1:100 dilutions, was included for SCoV2 RBD immunoblot (**G**). All immunoblot photographs were adjusted to 12% brightness and 5% contrast for consistency. (**H**) Lastly, the PBMC of transiently and chronically FCoV1-infected cats (4GC, G-3) were stimulated with either FCoV1 (red), FCoV2 (blue), or SCoV2 (green) RBD. The PBMC of two SPF cat were similarily stimulated with the RBDs. The expression of IFNγ of each stimulated PBMC is presented as the relative transcription of IFNγ mRNA level, relative to the GAPDH mRNA level, using qRT-PCR. The stimulation of T-cell mitogen ConA served as a positive control, whereas the RPMI media served as a negative control. The 1:100* is the serum dilution for cat J2d*, and cat J2d without asterisk is 1:400 in (**F**). The 1:50* is the serum dilution for only cat J2-4*, and cat J2-4 without asterisk is 1:1000 in (**G**).

## Data Availability

The RBD sequence data of FCoV1 RBD, FCoV2 RBD, CCoV1 RBD, CCoV2 RBD, human codon-optimized FCoV1 RBD, human codon-optimized FCoV2 RBD, and human codon-optimized SCoV2 UF2-RBD presented in this study are deposited and openly available in the NCBI database with accession numbers OP597272, OP597273, OP597274, OP597275, OP597277, OP597278, and OP597279, respectively. The complete sequences of the spike proteins used for our analyses, including their NCBI accession number, are shown in Figure 2 and Appendix A.

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
