# Peer review of "Both Feline Coronavirus Serotypes 1 and 2 Infected Domestic Cats Develop Cross-Reactive Antibodies to SARS-CoV-2 Receptor Binding Domain: Its Implication to Pan-CoV Vaccine Development"

_viruses, 2023, doi:10.3390/v15040914_

Round 1
Reviewer 1 Report
In this manuscript the authors describe the results of their studies to support the finding that FCoV type 1 and 2 infected cats develop cross-reactive antibodies to SARS-CoV-2 receptor binding domain. In most seroprevalence studies, initiated since the first description of SARS-CoV-2 infections, no clear indication of the induction of cross-reactive antibodies was found or only in incidental cases. And with the exception of one study by Hancock et al as discussed by the authors. The current study was conducted after the unexpected finding of antibodies in spf laboratory tom cats that were in contact with FCoV-positive queens during mating. To support and to confirm the finding of cross-reactivity with SARS-CoV2 RBD several studies were conducted. In general the methods used and the results of these studies are clearly described and discussed. This also includes the approach to identify the potential FCoV1 and FCoV2 RBDs. These findings are extensively discussed together with data from literature to substantiate the significance of these findings for the development of Pan-CoV vaccines.
The presentation and interpretation of the results do raise a number of questions that need critical evaluation also because some seem unexpected, difficult to understand and even contradictory.
Major and minor comments
- - In the M&M section it is mentioned that sera were screened for antibodies in an ELISA against FCoV-WV and the RBD of SCoV2 and FCoV RBD. But, only the ELISA results with FCoV-WV and SCoV2 RBD are shown and discussed. It is interesting and even important to show the reaction of the different sera against the (proposed) FCoV RBDs in the ELISA as well (and not just in the immunoblot). Coating with the whole virus preparation and the SARS-CoV2 is done with the same amount of protein. Hence the FCoV-WV preparation contains a much lower number of FCoV RBD molecules. When looking at the data of the stringent ELISA, it is also relevant to know the reactivity against the FCoV RBDs. Only one of the tom cats at 9 months post mating is considered FCoV positive(HOGT3: although weakly positive, just at the tresshold value) whereas all become positive for SARS-CoV2. Do these FCoV WV negative sera also react with the FCoV RBDs? The conclusion that the stringent ELISA confirmed that the sera from all three toms cross-reacted strongly with SCoV2 seems not correct since antibodies against FCoV could not be detected in the stringent ELISA. Cross-reaction implies that also antibodies against FCoV can be detected in the stringent ELISA.
- - The stringent ELISA was performed as a confirmation of the original finding with the overnight ELISA. In the stringent ELISA only one of the negative controls was included and the positive controls (UGA 1.4 and UGA 4.1) for FCoV and SCoV2 were not included at all. Although the sera of the queens showed reactivity against SCoV2 in the overnight ELISA these sera cannot be considered controls for the stringent confirmation ELISA. The results of these controls should therefore be included.
- - The information on how the cut-off values (or threshold values) in the ELISA were determined is lacking. The explanation of the red dotted lines (fig 1 D and E, presumably cut-off value?)) is also lacking in the legend of Fig 1.
- - There is a typo in line 366. March 2, 2022 should be 2020. Also it is debatable if the time difference between the blood collection (Jan 29, 2020) and the first report of COVID-19 cases (one month later) is a strong support for an FCoV infection and not SARS-CoV2 since it is very likely that infections occurred already before the original reporting.
- -The queens are presented as most likely infected with FCoV and a potential source of infection for the tom cats. This is based on serology but were the tom cats (e.g. when they had mild diarrhea) also tested for FCoV shedding by PCR?
- - The reactivity of the sera was tested in immunoblots with FCoV-WV, SCOV2, FCoV1 and FCoV2. The antibodies against SCoV2 seem to decline after infection in the queens as well as in the tom cats (fig 3A). This decline is also discussed by the authors. However, in the FCoV specific immunoblots not the same sera were tested as on the SCoV2 immunoblot strips and most often the sera from earlier time points after infection. For example, for 5HQT1 sera at 1,7 and 13 months are screened on SCoV2 immunoblot strips whereas on the others only serum taken at one month. It would make sense to screen the same sera tested on the SCoV2 RBD immunoblots also on the immunoblots containing the proposed FCoV 1 and 2 RBDs. This makes it more reliable to compare and to evaluate the results.
- - The supplementary fig S5 shows the results of immunoblots with sera of eight group housed laboratory cats. In these immunoblots the SCoV2 bands show a less distinct pattern as in the immunoblots presented in fig 3 and 4. The reaction of cat G-8 (S5B1) is not described (par 3.7) but seems similar to the reaction of serum of cat G-7. And both are not clearly visible. The sera of KM1 and KY are rated as being positive for SCoV2 but certainly for the result with KM1 this is doubtful (presented as a faint gray blur).
- - SCoV2 RBD was shown to block the FCoV2 infection of cells at a high, but non-toxic concentration. Toxicity seems to have been determined by staining of the cells. However, this does not exclude that cell metabolic activity is not reduced, which will affect viral replication. It would be favourable to measure toxicity by metabolic staining e.g. by using tetrazolium salts (e.g. MTT ).
- - A few typos: line 764: “”supports are our””, delete “”are””. Line 871, “origin” should be original
Reviewer 2 Report
The introduction is well referenced and provides a good amount of detail. It could be improved by more explicitly stating the goals/objectives of the study before giving additional background information. Owners may want to highlight that one aim was to identify the FCoV RBDs. My interpretation was that the authors wish to demonstrate experimentally that the that the RBD of both types of FCoV cross react with SARSCOV2 but that the RBD of FCoV 1 and 2 do not cross react with each other. (This is not surprising as their S1 subunits are quite distinct.)
Please clarify if the queens/Toms had been exposed to FCoV1 or 2 (line 31/32 suggests that cross reactivity was specific to FCoV1 RBD (line 318/319 state cats were positive for FCoV2 while lines 664/665 suggest the authors suspect the cats were infected with FCoV1.) FCoV is more prevalent naturally, and most cats in multi-cat environments are positive for FCoV, co-infections have been documented.
The methods are well documented. The results are well explained in the discussion section.
The authors detail the known receptors for most medically relevant coronaviruses, highlighting that the FCoV-1 receptor remains unknown. Do the offers have any inference as to the identity of the FCoV-1 receptor based on their results. Can the authors determine whether the cross reactivity is due to glycosylation versus linear AA’s by blocking glycosylation sites prior to incubating with sera?
FCoV-1 RBD blocking assays could be considered with cell culture adapted FIPV Black and Fcwf cells. FIPV-Black is adapted to use heparin sulfate for cell entry so it is not ideal, but at a minimum this should be addressed in the text. If performed, this experiment may provide interesting information about the nature of the cross reactivity.
